# Insight of Salt Spray Corrosion on Mechanical Properties of TA1-Al5052 Self-Piercing Riveted Joint

**DOI:** 10.3390/ma15238643

**Published:** 2022-12-04

**Authors:** Jiamei Lai, Zhichao Huang, Nanlin Tang, Zhaoxiao Hu, Yuqiang Jiang

**Affiliations:** 1Polymer Processing Research Laboratory, School of Advanced Manufacturing, Nanchang University, Nanchang 330031, China; 2Key Laboratory of Conveyance and Equipment, Ministry of Education, East China Jiaotong University, Nanchang 330013, China

**Keywords:** self-piercing riveting, salt spray corrosion, fatigue strength, fatigue failure form, titanium alloy

## Abstract

Self-piercing riveted (SPR) joints in automobiles inevitably suffer from corrosion damage and performance reduction. In this work, the influence of salt spray corrosion on the mechanical properties of TA1-Al5052 alloy SPR joints was studied. The TA1-5052 SPR joints were prepared and salt spray tests were carried out for different durations. The static and fatigue strengths of the joints after salt spray corrosion were tested to analyze the effect of salt spray duration on the performance of the joints. The results show that the joints’ static strength and fatigue strength decrease with prolonged salt spray time. The salt spray duration affects the joint’s tensile failure mode. The tensile failure without corrosion and with a short salt spray time is the fracture failure of the lower aluminum sheet, and the tensile failure of the joints after a long time of salt spray corrosion is the failure of the rivets. The fatigue failure form of the SPR joint is the formation of fatigue cracks in the lower aluminum sheet, and salt spray time has little effect on the fatigue failure form. Salt spray corrosion can promote the initiation and propagation of fatigue cracks. The fatigue crack initiation area is located at the boundary between the lower aluminum sheet and the rivet leg. The initiation of cracks originates from the wear zones among the sheet metal, rivets, and salt spray particles.

## 1. Introduction

Recently, lightweight technology has demonstrated broad application prospects in automobiles, and lightweight has become an inevitable trend in the future development of the automobile industry [1,2]. Low-density lightweight materials and high-strength materials are the main lightweight materials in automobiles, such as aluminum-magnesium alloys, titanium alloys, high-strength steels, and composite materials. These materials need to be connected. Self-Piercing Riveting (SPR) technology can effectively realize the connection of lightweight materials, and is widely used in production [3,4,5,6].

To realize the reliable connection of lightweight materials, many researches on SPR have been carried out. He et al. [7] researched the static strength of titanium alloy SPR joints, and the test results showed that the SPR joints with titanium alloys serving as upper sheets have higher static strength. Zhou et al. [8,9] analyzed the effects of rivet height and sheet lap sequence on the joint quality, and found that the shot peening process can improve the joint’s static and fatigue strength and reduce the fatigue crack expansion rate. Huang et al. [10] used SPR to realize the connection between the foam metal sandwich composite aluminum sheet and the aluminum alloy, and found that the foam sandwich reduced the maximum failure load in the tensile shear test, and increased the maximum failure displacement of the joint. Liu et al. [11] reported the joining mechanism and damage of SPR joints in carbon-fiber-reinforced polymer composites and aluminum alloy. In addition, when the strength difference between the joined sheets is relatively significant, the joints will fail on the material with poorer strength. Chung et al. [12] studied the fatigue properties of steel/aluminum SPR joints, and pointed out that the fatigue properties were affected by the strength of the lower sheet material. Zhang et al. [13] studied the fatigue and failure behaviors of the SPR joints of titanium sheets, and concluded that there is a direct relationship between the initiation of fatigue cracks and fretting wear. Karim et al. [14] studied the effect of riveting coatings on strength degradation between different connected materials. Çavuşoğlu et al. [15] found that the plate material and plate combination directly affect the performance of SPR joints. Neslušan et al. [16] and Skoczylas et al. [17] found that shot peening strengthens the alloys and joints.

In actuality, the body of automobiles is exposed to the atmosphere for a long time, which will result in corrosion damage and reduced performance. Calabrese et al. [18,19] systematically reported galvanic corrosion damage on steel-aluminum SPR joints, and their results showed that corrosion degradation largely affects the strength of hybrid joints. Fiore et al. [20] studied the effect of salt spray aging on the mechanical properties of SPR joints of aluminum alloy, glass fiber, and carbon fiber composite sheets. Huang et al. [21] compared the corrosion performance of SPFC440 steel/5052 aluminum SPR joints with glue riveting composite joints, and found that the presence of adhesive not only improved the joint strength, but also protected the metal interface and reduced the galvanic corrosion. Kharitonov et al. [22] studied the corrosion performance of AA6063-T5 aluminum alloy in NaCl solution containing molybdenum salt, and analyzed the corrosion inhibition mechanism of molybdate by electrochemical, microscopic, and spectroscopic experiments.

Although some researches on the connection and corrosion of SPR joints have been reported, there are few studies on corrosion and fatigue degeneration of dissimilar metal SPR joints, and the corrosion failure mechanism of SPR joints needs to be continuously explored. The relationships between the mechanical properties, failure mechanism, and corrosion degradation of titanium-aluminum SPR joints need to be further studied. In this study, industrial pure titanium TA1 sheet and 5052 aluminum alloy sheets were jointed, and the corrosion behaviors were analyzed. Then, the mechanical properties of the corroded samples were tested to analyze the effects of salt spray durations on joint performance and failure mechanism.

## 2. Materials and Methods

### 2.1. Materials

The alloys used in this work are commercial TA1 titanium alloy and 5052 aluminum alloy sheets. The chemical compositions and mechanical properties of TA1 and 5052 alloy sheets are shown in Table 1 and Table 2.

### 2.2. Self-Piercing Riveting Joints

According to the research of scholars, as well as the authors’ previous works, riveting parameters, such as sheet thickness and sheet lap sequence, have great influence on the performance of self-piercing riveting joints [23,24,25]. When different metal plates are connected, the more plastic plates should be placed on the lower layer; at the same time, the rivet height should not exceed the total thickness of the plate by 3 mm. The purpose of this paper is to investigate the effect of corrosion on the performance of the joint, so the selection of parameters is not overly discussed. Subsequent work has been based on better riveting results.

The dimensions of the TA1 and the 5052 alloy sheets were 110 mm × 20 mm × 1.5 mm. The size of the overlapping area was 20 mm × 20 mm. According to the thickness and mechanical properties of the alloy sheets, the rivet had a hardness of H4 and a height of 6 mm. The dimensions of the bottom die, rivets, and sheet overlap are shown in Figure 1.

### 2.3. Salt Spray Corrosion Test

There are many ways to evaluate corrosion, such as polarization testing, salt spray testing, etc. The purpose of this paper is to simulate the impression of corrosion on joint performance in an atmospheric environment. Salt spray corrosion (SSC) is more suitable for practical applications, therefore, SSC is used. To reveal the influence of SSC on the mechanical behaviors of the TA1/5052 joints, the riveted samples were brushed and washed with acetone to remove the surface grease, and then the samples were dried in a 70 °C incubator for 48 h. The samples were finally weighed.

The SSC tests of SPR joints were carried out according to the GB/T 10125-2012 [26]. The tests were executed at a temperature of 35 ± 1.5 °C; the corrosion environment was constructed using NaCl solution with a density of 50 ± 5 g/L; and the pH value of the NaCl solution was between 6.5 and 7.2. The durations of the SSC were 0 week (0 h), 1 week (168 h), 3 weeks (504 h), 5 weeks (840 h), and 7 weeks (1176 h), and five samples were used at each SSC time tests to minimize the errors. After the SSC tests, the samples are dried for 30 min at room temperature, and then the samples were rinsed with running water for 30 min to remove the residual salt spray solution on the surface of the samples. The samples were dried at 70 °C for 48 h, and finally, the samples were weighed.

### 2.4. Mechanical Property Test and Fracture Morphology

The tensile-shear test was carried out on an RGM 4030 testing machine, and the 20 mm × 20 mm × 1.5 mm gaskets were supported on both ends of the samples to overcome the torque of the specimen. Each group of samples was subjected to three repetitive tensile experiments. The tensile rate was 2 mm/s, and the average static failure load of the tensile samples was referred in the subsequent fatigue tests. In the fatigue test, 60% of the maximum static load was selected as the maximum fatigue load, so the magnitude of the cycles was distributed in 10^4^–10^6^. Stress ratio R = 0.1. The fracture morphology was photographed by Hitachi SU8010 scanning electron microscope (SEM, Tokyo, Japan). The microscopic morphology of the fracture was used to analyze the failure mechanism.

## 3. Results and Discussion

### 3.1. Forming Quality

Generally, the cross-sectional geometry and the corresponding quality parameters, such as the inner lock length *u*, the inner lock height *u_h_*, and the residual bottom thickness *t_r_*, are the key indicators to evaluate the forming quality of the SPR joints [27]. Figure 2 is the cross-section and characteristic dimensions of the joint after forming. It can be found from Figure 2 that the internal lock length of the TA1-5052 SPR joint is 1.4518 mm, the internal lock height is 3.1160 mm. Du et al. [28] investigated the forming quality of the joints, and suitable forming parameters were reported. According to the results, the SPR joints obtained in this experiment have good forming quality.

### 3.2. Corrosion Mechanism and Morphological Analysis

In the SSC test of TA1 titanium plate and 5052 aluminum alloy plate SPR specimens, the types of corrosion occurring are: pitting corrosion of the sheet metal; galvanic coupling corrosion between the plate and rivet; and crevice corrosion of the lap jointed area. Among them, pitting corrosion of 5052 aluminum alloy plate is the most common. Pitting corrosion generally sprouts in the metal surface passivation film (Al_2_O_3_) defects and mechanical damage parts [29,30,31]. This is because the solution of corrosive anions (Cl^−^) adsorbed in the passivation film defects, and the cations (Al^3+^) combine to generate soluble chloride (AlCl_3_), thus forming a small pitting holes. After the pitting hole is created, the inside of the hole is in an activated dissolved state and the outside of the hole is in a passivated state, thus constituting an activated passivated corrosion cell and promoting the development of pitting damage.

Figure 3 shows the corrosion morphology of the joint before and after salt spray. During the salt spray aging process, a large amount of white rust (AlCl_3_) was formed on the surface of the aluminum plate and the area around the rivet of the joint, while there were no obvious rust stains on the surface of the titanium plate, indicating that the corrosion on the aluminum side was much more serious than that on the titanium side. Without salt spray corrosion of the specimen, the upper and lower metal plate presented no rust stains, no obvious defects at the joints. After 7 weeks of SSC, the sample was dull, the aluminum plate was almost completely covered by white rust, a small amount of rust stains on the surface of the titanium plate, most of the zinc coating on the rivet surface was dissolved, and there were a large number of brown rust stains on the rivet head; the aluminum plate and the “rivet buckle” at the edge of the lap joint area were seriously corroded, and corrosion pits were formed on the surface (Figure 3b, red circles). Some of the white rust was corroded and flaked off, and the surface roughness of the specimen increased.

### 3.3. Corrosion Resistance

The corrosion rate of metal is a key parameter by which to express the corrosion resistance of the alloys and their joints [32,33,34]. In the case of uniform corrosion and proper removal of corrosion products from the metal surface, the corrosion rate is expressed as Equation (1).
(1)$${}^{K^{+}\;=\;\frac{g_{0}-g_{1}}{s_{0}\cdot t}}$$
where *K*^+^ is the corrosion rate (g/m^2.^h), *g*_0_ is the sample quality before corrosion (unit: g), *g*_1_ stands for the sample quality after corrosion (unit: g), *S*_0_ indicates the surface area of the sample (unit: m^2^), and *t* is the corrosion time (h).

The samples with different SSC durations are weighed, five samples of the same SSC durations are weighed, and the average mass loss is calculated. Then, the average corrosion rate of the samples is obtained according to Equation (1), as shown in Figure 4. The dot and red line are the experimental and predicted average corrosion rates, respectively.

The cubic polynomial can be used to describe the variation tendency between the average corrosion rate and corrosion duration of each sample, as shown in Figure 4 and Equation (2):(2)$$K^{+}=1.197t^{3}-2.042t^{2}+1.180t+0.002$$

As shown in Equation (2) and Figure 4, the average corrosion rate of the samples increases with the increased corrosion time. At the beginning of the SSC durations, the average corrosion rate quickly increases; then, the steady average corrosion rate is obtained. The average corrosion rate is quickly increased at the end of the SSC durations. Due to the high chemical activity of the aluminum element, a dense Al_2_O_3_ film is formed on the surface of the aluminum sheet. The existence of the Al_2_O_3_ film on the surface of the aluminum sheet avoids the direct exposure of the aluminum sheet to the SSC environment, which reduces the erosion of the aluminum sheet by the salt spray particles [35,36,37]. At the same time, the Al_2_O_3_ film separates the aluminum sheet and the titanium sheet, which reduces the electrochemical corrosion caused by the direct contact between the two connected sheets. Therefore, the average corrosion rates of the samples corroded for 1 week and 3 weeks were smaller, as shown in Figure 4. After 5 weeks of corrosion, the Al_2_O_3_ film on the surface of the aluminum sheet is degraded and failed due to the erosion of salt spray particles. The salt spray particles are directly adhered to the surface of the aluminum sheet, which accelerates the corrosion of the aluminum sheet. Moreover, the salt spray particles that are attached to the SPR joints increase the crevice corrosion and electrochemical corrosion, resulting in the accelerated corrosion rate of the samples [38]. This phenomenon is more pronounced after 7 weeks of corrosion, as shown in Figure 4.

### 3.4. Tensile Static Strength

Figure 5 shows the load-displacement curves and the energy absorption values of joints under different corrosion durations. Compared with the uncorroded joint, the maximum static load of the joint after 1 week of corrosion has no significant effect, as shown in Figure 5a. Generally, the titanium sheet has strong corrosion resistance, and the Al_2_O_3_ film on the surface of the aluminum sheet blocks the erosion of the aluminum sheet matrix by the salt spray particles [39]. Therefore, the salt spray has less influence on the tensile properties of the joint that suffered from 1 week of corrosion; shown in Figure 5a. After 3 weeks of corrosion, the maximum static failure load of the joints decreases with the prolonged corrosion time. This is due to the degradation failure of the Al_2_O_3_ film and the coating on the rivet surface, which accelerates the galvanic coupling corrosion between the riveted sheets, and finally degrades the riveted joint performance. At the same time, the corrosion products accumulate at the riveted area, and act as a lubricant during the stretching process, which accelerates the failure of the joints. Therefore, the strength of the joint decreases, and the maximum static failure load continuously decreases with the prolonged SSC time; as shown in Figure 5a. When the SPR joints corrode for 7 weeks, the joints are severely damaged, and the maximum static failure load is sharply decreased.

The area surrounded by the x-y axis represents the energy absorption values of the joints [40], and Figure 5b shows the corresponding values of the joints under different corrosion durations. Compared with the uncorroded joints, the energy absorption values are reduced by 24.84%, 44.93%, 57.60%, and 72.59% after 1, 3, 5, and 7 weeks of SSC tests, respectively. It can be seen that the ability of the corroded joint to withstand the action of external forces is significantly weakened.

### 3.5. Tensile Failure

#### 3.5.1. Macroscopic Analysis of Tensile Failure

Figure 6 shows the tensile failure forms of joints with various SSC duration, and the reproducible morphologies are simultaneously expressed. The SSC duration affects the tensile failure form of the joints (Figure 6). The failure mode of the uncorroded joint is caused by the fracture of the lower sheet, as shown in Figure 6a. During the stretching process, the two sheets are gradually bent and deformed by the tensile load. Due to the good interlocking performance of the joint, the maximum load that the joint can withstand is higher than the yield strength of the aluminum sheet. When the tensile load exceeds the yield strength of the aluminum sheet, the lower aluminum sheet breaks, the joint fails, and the maximum static load rapidly decreases to zero; shown in Figure 5a. The failure modes of the joints after 1 week and 3 weeks of SSC are the same as that of the uncorroded samples, both of which are the failure of the lower aluminum sheet, and the rivets remain on the upper sheet, as shown in Figure 6b,c. The two sheets are plastically deformed for the joints corroded for 1 week and 3 weeks. However, the bending degree of the sheet decreases with the increased SSC time, which is caused by the weakened plastic properties of the sheets after SSC. After 5 and 7 weeks of SSC, the joints show that the failure of the rivet, the rivet tail, and the lower aluminum sheet are separated, and the rivet is pulled out from the lower sheet, as shown in Figure 6d,e. Combined with Figure 4 and Figure 5, it can be seen that, after a long time of SSC, the joints are affected by the galvanic corrosion, the pitting corrosion of the aluminum sheet, and crevice corrosion, resulting in a significant decrease in static performance. Therefore, the maximum load the joint can bear is lower than the yield strength of the aluminum sheet. During tension deformation, when the tensile load exceeds the maximum load the joint can withstand, the severe deformation of the two sheets causes the separation of the rivet from the lower aluminum sheet.

#### 3.5.2. Microscopic Analysis of Tensile Failure

Figure 7 shows the microscopic fracture morphologies of the uncorroded joint. The crack on the lower aluminum sheet propagates from one side of the sheet to the other side along the inner wall of the die, and the crack propagation direction is perpendicular to the load direction, as shown in Figure 7a. The micro-morphologies of the left and right sides of the aluminum sheet are shown in Figure 7b,d. The fracture presents a typical dimple morphology, and the dimple is convex toward the end of the crack propagation direction. Due to the good toughness and deformation ability of the aluminum alloy, the tensile plastic deformation occurs in the aluminum alloy in the tensile process. When the plastic deformation is increased to a critical value, the dislocation density sharply increases, leading to the formation of micro-cracks at the grain boundaries and second phases. These micro-cracks are gradually grown up, and eventually cause the break of the sheet. However, as the crack propagates, the depth of the dimple in Figure 7d is shallower than that in Figure 7b, which is attributed to the large deformation of area C in the final failure stage of the joins. However, area B is the inner wall area of the concave die, where there is no intense tearing and only the separation of the rivet from the lower aluminum plate causes the metal delamination phenomenon [41]. Therefore, unlike in Figure 7b,d, no dimples are observed in Figure 7c.

Figure 8 shows the micro morphologies of the joint after 7 weeks of corrosion. It can be seen from Figure 8a that the tail of the rivet falls off from the lower aluminum sheet, and the rivet pulls out from the lower sheet. There is a large crack perpendicular to the load direction on the aluminum alloy sheet, and the large crack is in a location near the die. Due to the long-term SSC of the aluminum sheet, and the stress concentration at the bottom of the aluminum sheet, many corrosion pits remain at the bottom of the aluminum sheet. In addition, when the aluminum sheet is corroded, the performance of the sheet is weakened. Under the tensile load, the corrosion pit area of the aluminum alloy is prone to damage and failure [42]. Figure 8b shows the enlarged view of area A, and there is a wear area left on the inner wall of the aluminum sheet. This is caused by the friction and scratching between the rivet and the lower aluminum sheet in the pulling out of the rivet from the lower sheet. Figure 8c is an enlarged view of area 1, and a smooth and flat cross-section can be found. It indicates that the frictional effect between the rivet and the lower aluminum plate is weakened in the pull-out process. Figure 8d is an enlarged view of area 2. It can be seen that the metal peeling phenomena and the residual multiple wear impurities are presented, showing the regular longitudinal groove pattern. In addition, the spherical material on the fracture surface is NaCl crystals attached to the surface of the aluminum plate. Due to the long time in the salt spray environment, the salt spray particles will gradually erode into the joint gap, and the presence of corrosion pits also helps the attachment of salt spray particles. The authors have verified this conclusion by EDS in the previous work [21].

### 3.6. Fatigue Strength

Usually, the load level is often used instead of the stress level in the fatigue test research of SPR joints, and the fatigue life is explored through the F-N curve. The load-life (F-N) curves of the samples at different SSC durations are obtained, as shown in Figure 9, and the corresponding fitted equation is shown in Table 3.

According to the F-N data, the following model can accurately describe the relationships among the maximum load, corrosion time, and cycles; Equation (3):(3)$$F=(0.0011t^{2}+0.5510t+101.5754)\mathrm{lg}N^{\left(7.5070\times 10^{-7}t^{2}-0.0027t-2.5154\right)}$$
where *F* is the maximum load (kN), *N* represents the cycles (kC), and *t* is the corrosion time (h).

For the given corrosion duration, the fatigue life of the joint decreases with the enhanced maximum load. The fatigue failure of the SPR joints is caused by the continuous accumulation of microscopic losses in the joint. The greater the cyclic stress, the more serious the slip deformation of the surface metal, the easier the formation of the slip band, and the earlier the initiation of fatigue cracks [43]. Moreover, the crack propagation speed is affected by the level of the load. The larger the load, the faster the fatigue crack propagation speed, and the more likely the joint fails.

On the other hand, the fatigue life of the joints is decreased with the increase of SSC duration, and the F-N curves show a downward trend. The salt spray particles that are attached to the joint gap will aggravate the fretting wear of the joint, and reduce the internal bearing area of the joint, therefore, promoting the initiation of fatigue cracks. SSC causes the pitting corrosion of the lower aluminum sheet, and the stress concentration occurs in the corrosion pits of the aluminum sheet surface, which further aggravates the initiation and expansion of fatigue cracks. Moreover, long-term SSC can affect the performance of the material. The longer the SSC time, the worse the material performance and the easier the failure of the joints.

### 3.7. Fatigue Failure

#### 3.7.1. Macroscopic Analysis of Fatigue Failure

Figure 10 shows the fatigue failure modes of joints at different SSC durations. The percentage marked in the figure is the ratio of the applied max load in the fatigue tests and the max static load of the joints. It can be seen from Figure 10 that the SSC duration does not affect the fatigue failure mode of the joint.

The fatigue failures of the joints at different SSC durations are all due to the formation of the fatigue cracks in the lower aluminum sheet, which cause the failure of the joints. The fatigue crack spreads from the riveted hole to the left and right edges of the aluminum sheet, and the fracture direction is perpendicular to the load direction. The reason is that there is residual stress at the junction of the rivet foot and the aluminum sheet during the riveting process, where the stress concentration is significant. Under cyclic loading, the stress concentration area is prone to plastic deformation, and the repeated cyclic plastic strain causes the extrusion and intrusion of the crystalline metal phase, resulting in slip bands and the final formation of microcrack nuclei [44]. Once the micro-crack is nucleated, the micro-crack will expand and extend along the slip plane, and the micro-crack will gradually develop into a macro-crack. At this time, the crack propagation direction is perpendicular to the load direction. Due to the prolonged stress cycles, the size of the crack continues to expand. When the crack expands to a critical size, the stress concentration reaches the fracture strength of the material, resulting in the final instantaneous fracture of the sample.

#### 3.7.2. Microscopic Analysis of Fatigue Failure

Figure 11 shows the fatigue fracture micromorphology of uncorroded joints. There are apparent wear marks in the rivet area at the junction of the two sheets, indicating that severe fretting wear has occurred due to cyclic load.

Figure 11b shows that there is a fatigue crack about 0.2 mm long at the junction of the lower sheet and the rivet foot (area 1) because the riveting process will cause stress concentration at the junction between the lower sheet and the rivet foot. Slip bands are easily generated under cyclic loading, resulting in the formation of microcracks. The continuous expansion of microcracks eventually generates fatigue cracks. Area 1 is magnified and observed, as shown in Figure 11c, the fracture presents a dimple characteristic. Area 2 is the fatigue crack propagation zone, and the magnified morphology is shown in Figure 11d. This region presents a ridge-like morphology, which is due to the propagation of different microcracks along different crystal planes, and the cleavage fracture morphology is finally formed [45,46]. In area 3, there is a layered step-like morphology, some micro-cracks continue to expand to the surrounding area, this area is the fatigue micro-crack initiation area. The metal in the crack source area is squeezed in and out, resulting in many microscopic cavities, as shown in Figure 11e.

Figure 12 shows the fatigue fracture morphology of the joint after 7 weeks of corrosion at 50% load. It can be seen that the fatigue micro-cracks initiation area is located at the junction of the lower aluminum sheet and the upper sheet. This area becomes a vulnerable area due to the stress concentration caused by the corrosion pit. Furthermore, the adhesion of salt spray particles accelerates the fretting wear, and eventually lead to microcracks generation. Area 1 in Figure 12b is the fatigue crack propagation area, and the enlarged observation of this area is shown in Figure 12c. There is a river-like appearance, which is caused by the continuous intersection of numerous micro-crack propagations in the crack source area. There is a deep fatigue crack at the junction of the rivet foot and the lower sheet (area 2), because of the high incidence of micro-motion wear between the rivet foot and the sheet. Magnified observation of area 2 is shown in Figure 12d, the wear and spalling that appear around the cracks can be found, and some impurity particles are present.

## 4. Conclusions

In this work, the relationships between the mechanical properties, the failure mechanism, and the corrosion degradation of titanium-aluminum SPR joints were studied, and the following conclusions were made:(1)The average corrosion rate of the samples increased with the increase of corrosion time. At the beginning of the SSC durations, the average corrosion rate quickly increased, and then the steady average corrosion rate was obtained. The average corrosion rate quickly increased at the end of the SSC durations.(2)The static strength and fatigue strength of the joint decreased with prolonged SSC time. The SSC time affected the tensile failure mode of the joint but had no noticeable effect on the fatigue failure mode of the joint. The tensile failure without corrosion and with a short SSC time was the fracture failure of the lower aluminum sheet, and the tensile failure of the joints after a long SSC time was the failure of the rivets. The fatigue failure form of the SPR joint was the formation of fatigue cracks in the lower aluminum sheet.(3)SSC promoted the initiation and propagation of fatigue cracks in joints. The fatigue cracks of the joints before corrosion were initiated in the junction area of the two sheets and the boundary between the lower aluminum sheet and the rivet legs. After corrosion, the joints generally had multiple fatigue crack initiation areas.

## Figures and Tables

**Figure 1 materials-15-08643-f001:**
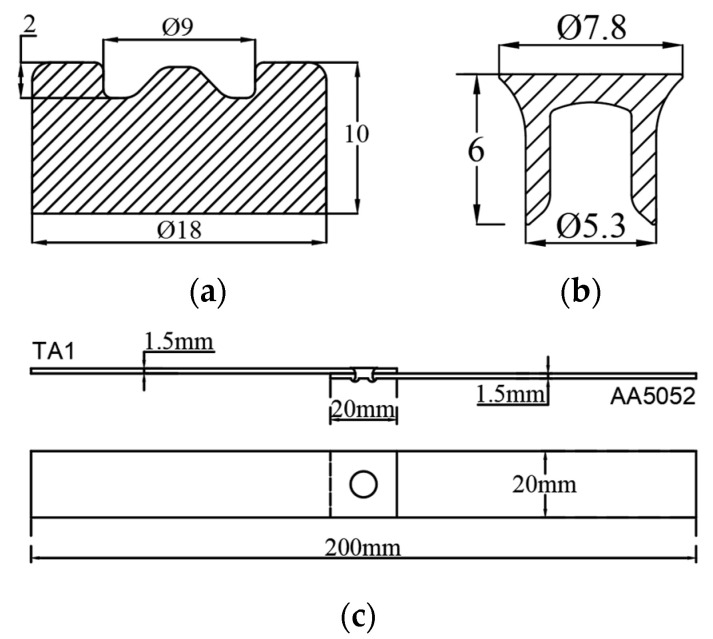
The geometric dimensions of (**a**) bottom die; (**b**) rivet; (**c**) lap dimensions.

**Figure 2 materials-15-08643-f002:**
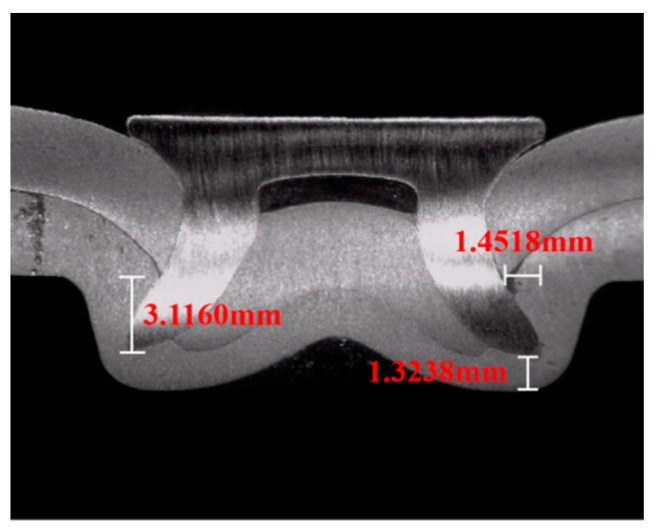
Cross-section and characteristic dimensions.

**Figure 3 materials-15-08643-f003:**
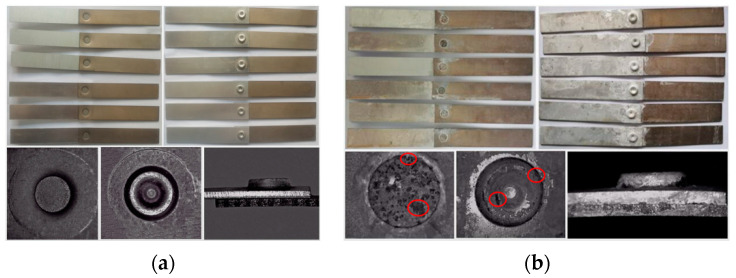
Appearance of the joints at different corrosion times of (**a**) 0 weeks; (**b**) 7 weeks.

**Figure 4 materials-15-08643-f004:**
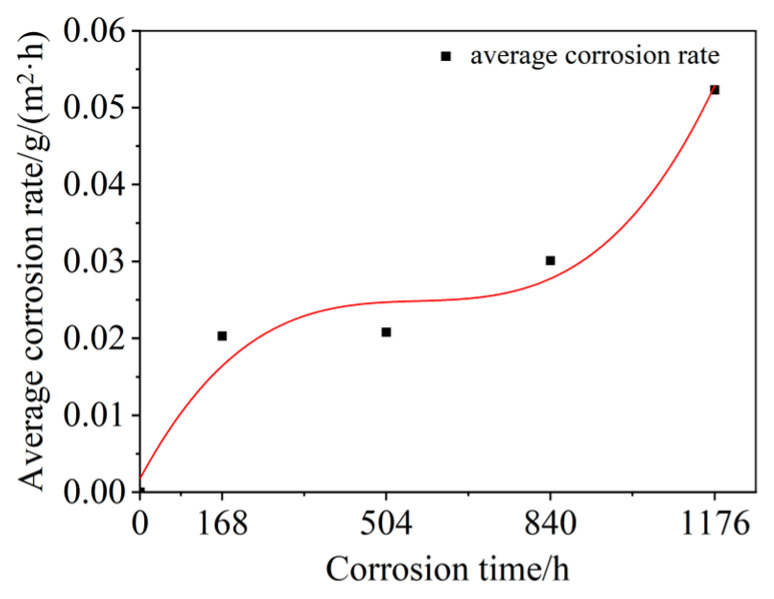
Corrosion rate of the SPR joints with different SSC durations.

**Figure 5 materials-15-08643-f005:**
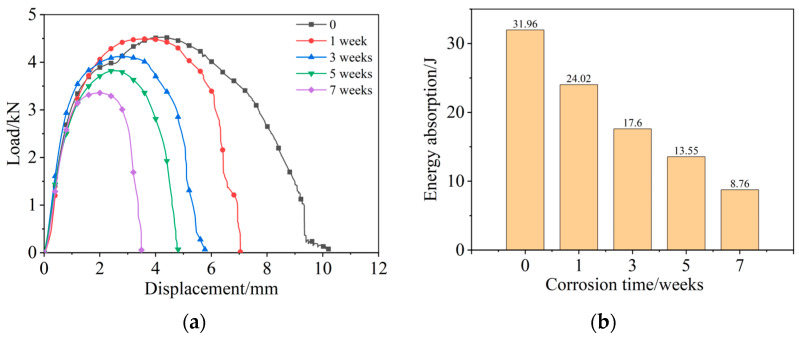
Load-displacement curves (**a**) and energy absorption values (**b**) of joints under different corrosion durations.

**Figure 6 materials-15-08643-f006:**
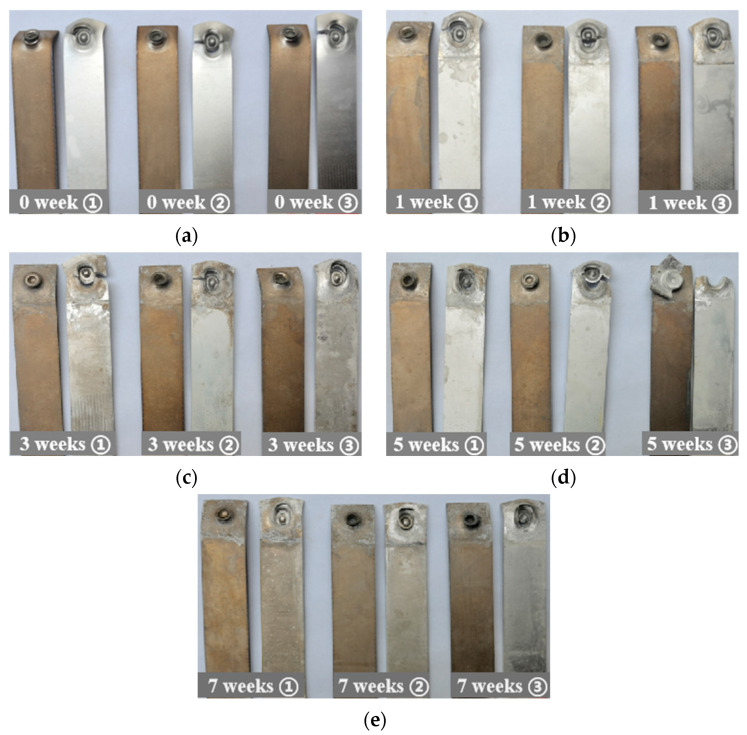
Tensile failure forms of joints at different corrosion times of (**a**) 0 week; (**b**) 1 week; (**c**) 3 weeks; (**d**) 5 weeks; and (**e**) 7 weeks.

**Figure 7 materials-15-08643-f007:**
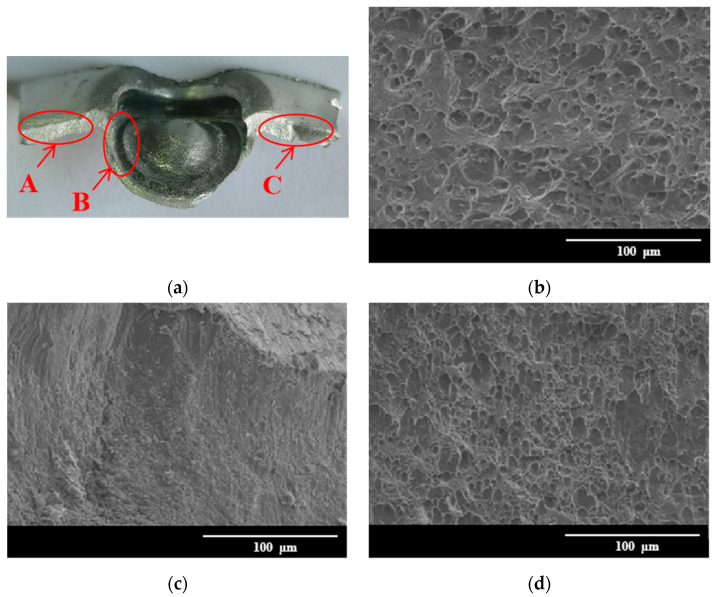
Tensile fracture morphologies of uncorroded joints: (**a**) macroscopic morphology of fracture; (**b**) enlarged view of area A; (**c**) enlarged view of area B; (**d**) enlarged view of area C.

**Figure 8 materials-15-08643-f008:**
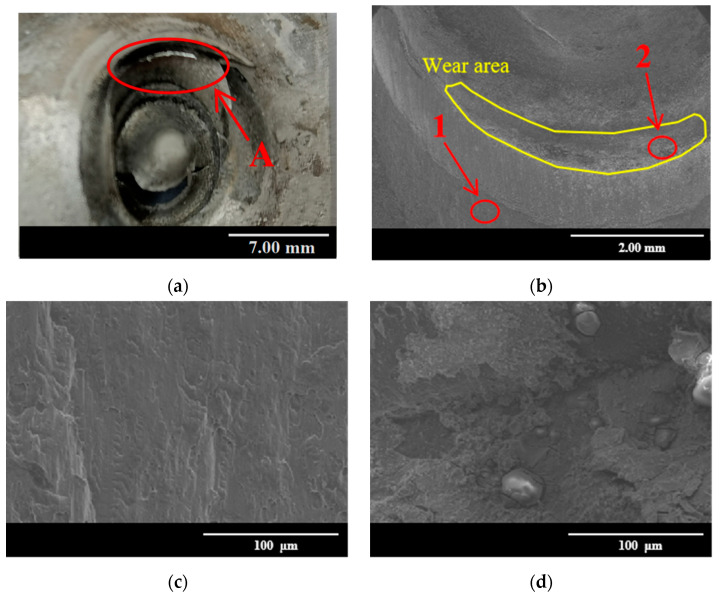
Tensile fracture morphologies of joints after 7 weeks of corrosion: (**a**) macroscopic morphology; (**b**) enlarged view of area A; (**c**) enlarged view of area 1; (**d**) enlarged view of area 2.

**Figure 9 materials-15-08643-f009:**
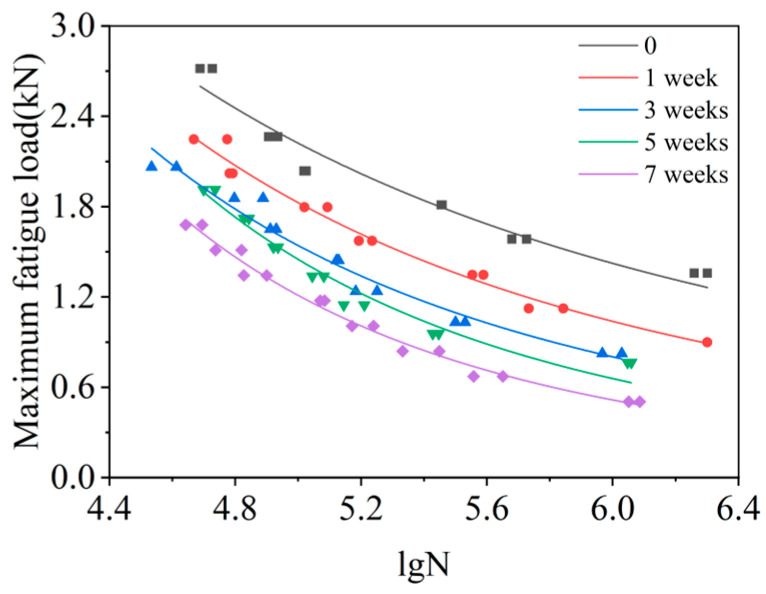
Load-life (F-N) curves of the SPR joints at different SSC durations.

**Figure 10 materials-15-08643-f010:**
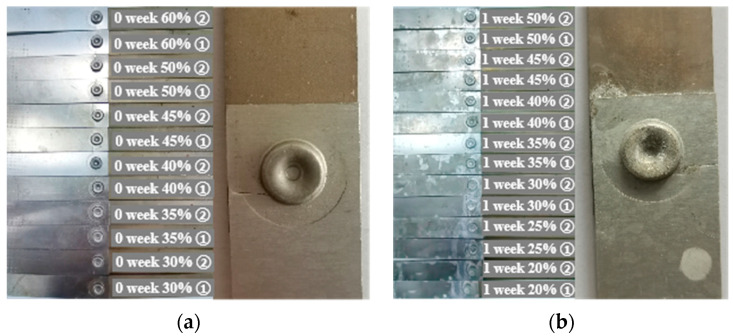
Fatigue failures of the SPR joints at different corrosion durations: (**a**) 0 week; (**b**) 1 week; (**c**) 3 weeks; (**d**) 5 weeks; (**e**) 7 weeks.

**Figure 11 materials-15-08643-f011:**
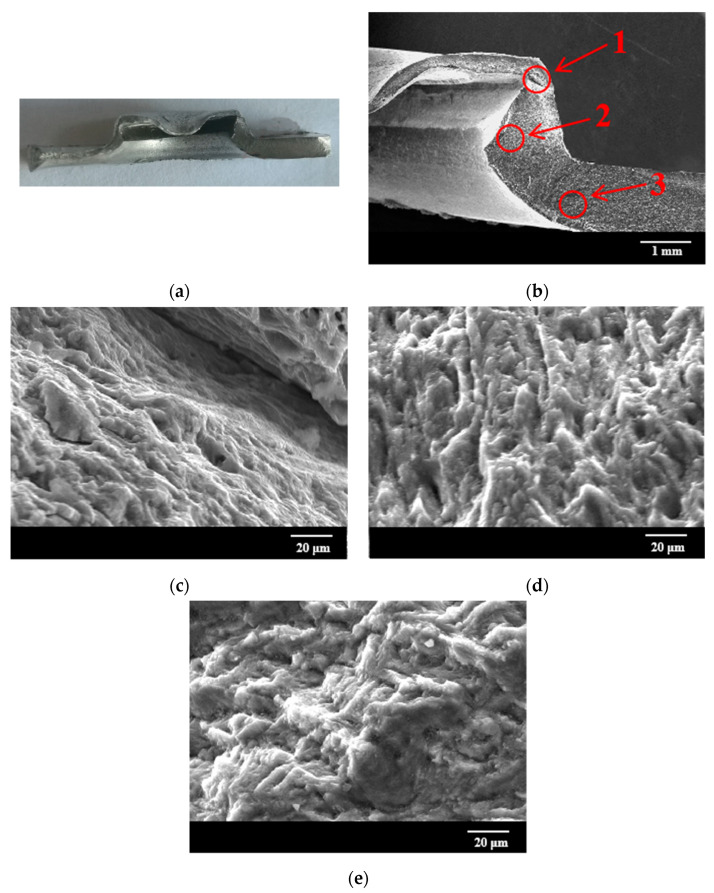
Fatigue fracture micromorphology of uncorroded joints: (**a**) macroscopic morphology; (**b**) microscopic morphology; (**c**) 800× magnification of area 1; (**d**) 800× magnification of area 2; (**e**) 800× magnification of area 3.

**Figure 12 materials-15-08643-f012:**
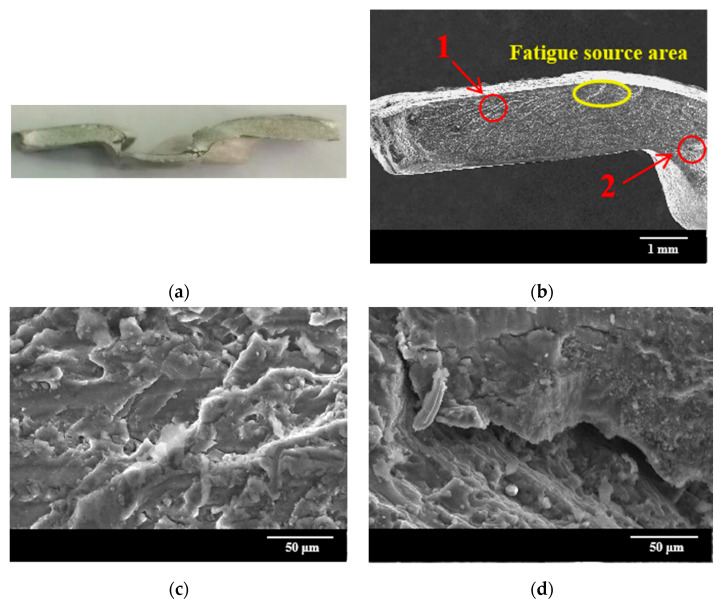
Fatigue fracture micromorphology of joints after 7 weeks of corrosion under 50% load: (**a**) macroscopic morphology; (**b**) microscopic morphology; (**c**) 500× magnification of area 1; (**d**) 500× magnification of area 2.

**Table 1 materials-15-08643-t001:** Chemical compositions of TA1 and 5052 alloys.

TA1	Element	Fe	C	N	H	O	Ti		
	Standard value	≤0.25	≤0.10	≤0.03	≤0.015	≤0.15	Bal.		
	Measured value	0.019	0.015	0.0098	0.0011	0.055	Bal.		
5052	Element	Si	Cu	Mg	Zn	Mn	Cr	Fe	Al
	Standard value	≤0.25	≤0.1	2.2~2.8	≤0.1	≤0.1	0.15~0.35	≤0.4	Bal.
	Measured value	0.072	0.006	2.627	0.018	0.03	0.1719	0.173	Bal.

**Table 2 materials-15-08643-t002:** Mechanical properties of TA1 and 5052 alloy sheets.

Materials		Tensile Strength (MPa)	Yield Strength (MPa)	Elongation (%)
TA1	Standard value	≥240	240~310	≥30
	Measured value	324	275	55
5052	Standard value	210–260	≥130	≥6
	Measured value	229	173	≥14

**Table 3 materials-15-08643-t003:** F-N curve equation of the joints at different SSC durations.

SSC Duration	F-N Curve Equation	r^2^
0	F = 113.06 (lg*N*)^−2.4418^	0.9516
1 week	F = 268.09 (lg*N*)^−3.0999^	0.9804
3 weeks	F = 490.33 (lg*N*)^−3.5805^	0.9583
5 weeks	F = 1548.97 (lg*N*)^−4.3333^	0.9624
7 weeks	F = 2250.48 (lg*N*)^−4.6773^	0.9826

## Data Availability

Not applicable.

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
