# Peer review of "Insight of Salt Spray Corrosion on Mechanical Properties of TA1-Al5052 Self-Piercing Riveted Joint"

_materials, 2022, doi:10.3390/ma15238643_

Round 1
Reviewer 1 Report
The manuscript "Insight of salt spray corrosion on mechanical properties of TA1-Al5052 self-piercing riveted joint" deals with some interesting aspects of the relationships between the mechanical properties, failure mechanism, and corrosion degradation of titanium-aluminum SPR joints. The authors have very nicely represented the work with attractive images. The microstructure analysis, corrosion analysis and mechanical properties assessments have been comprehensively investigated. I strongly recommend this manuscript for publication, as this manuscript also addresses a practical issue. The manuscript, however, requires some revisions before being considered for publication.
1. I wonder if authors have done any corrosion test (like polarization test) or all the reported corrosion data is from the formula 1:
2. Authors have mentioned that “The existence of the Al2O3 film on the surface of the aluminium sheet avoids the direct exposure of the aluminium sheet to the SSC environment, which reduces the erosion of the aluminium sheet by the salt spray particles.” I wonder if you can provide an image or any kind of support for this argument.
3. Please provide the chemical composition and mechanical properties of your alloys. What you have given in Tables 1 and 2 seems to be standard values. Please provide data from this specific (together with standard values).
4. While the fracture modes in Figures 6b and 6d are very well explained, there is hardly any explanation on the fracture surface, in Figure 6c. In this figure, dimples are not observed. Please explain
5. Please add scale bar to Figure 7a.
6. In Figure 7d, what exactly are those spherical features on the fracture surface? Do you have any EDS analysis? If so, please add it to the manuscript.
7. Authors have argued that the fatigue micro-cracks initiation area is located at the junction of the lower aluminum sheet and the upper sheet and that this area becomes a vulnerable area due to the stress concentration caused by the corrosion pit. I wonder if you can provide images of mentioned pits.
8. I strongly suggest authors give a short overview of different possible wear and corrosion mechanisms, maybe in a short paragraph. This is really missing in the manuscript. The current structure gives the feeling that authors first have introduced the problem and then sort of moved to the results. I think for readers it would be helpful to have a short overview of corrosion mechanisms perhaps somewhere at the end of introduction. I strongly suggest authors check and refer to the following articles. Though the topic might not be directly related to your article, but together they provide explanations for different corrosion mechanisms.
- Fan, B., Zhao, X., Liu, Z., Xiang, Y., & Zheng, X. (2022). Inter-component synergetic corrosion inhibition mechanism of Passiflora edulia Sims shell extract for mild steel in pickling solution: Experimental, DFT and reactive dynamics investigations. Sustainable Chemistry and Pharmacy, 29, 100821. doi: https://doi.org/10.1016/j.scp.2022.100821
- Xie, J., Zhang, J., Zhang, Z., Yang, Q., Guan, K., He, Y., Wu, R. (2022). New insights on the different corrosion mechanisms of Mg alloys with solute-enriched stacking faults or long period stacking ordered phase. Corrosion science, 198, 1. doi: 10.1016/j.corsci.2022.110163
- Gong, P., Wang, D., Zhang, C., Wang, Y., Jamili-Shirvan, Z., Yao, K., Wang, X., Corrosion behavior of TiZrHfBeCu(Ni) high-entropy bulk metallic glasses in 3.5 wt. % NaCl. npj Mater Degrad 6, 77 (2022). https://doi.org/10.1038/s41529-022-00287-5
12. The conclusion is very short and does not fully reflect the findings of your article. Please present more details in your conclusion.
Author Response
Ref: Materials-2066541
Title: Insight of salt spray corrosion on mechanical properties of TA1-Al5052 self-piercing riveted joint
Journal: Materials
Dear reviewers and editor:
Thanks for the suggestive advice from reviewers and editors. The manuscript has been carefully checked again. The following are the response to the comments from the reviewers and editor. The corresponding modifications have been made and the important corrections are highlighted in YELLOW in the revised manuscript.
Comments from the reviewers:
Reviewer 1#: The manuscript "Insight of salt spray corrosion on mechanical properties of TA1-Al5052 self-piercing riveted joint" deals with some interesting aspects of the relationships between the mechanical properties, failure mechanism, and corrosion degradation of titanium-aluminum SPR joints. The authors have very nicely represented the work with attractive images. The microstructure analysis, corrosion analysis and mechanical properties assessments have been comprehensively investigated. I strongly recommend this manuscript for publication, as this manuscript also addresses a practical issue. The manuscript, however, requires some revisions before being considered for publication.
- I wonder if authors have done any corrosion test (like polarization test) or all the reported corrosion data is from the formula 1:
Response: Thanks. Generally, there are many ways to evaluate the corrosion behaviors of the SPR joints, such as polarization testing, salt spray testing, exfoliation corrosion, etc. The salt spray corrosion is widely used for the evaluation of anti-corrosion behaviors of the structure in practical applications. The effect of corrosion on joint performance is studied using salt spray corrosion. The experimental procedure of the salt spray corrosion test is described in section 2.3. Salt spray corrosion test (Pages 3 of the revised manuscript).
- Authors have mentioned that “The existence of the Al2O3 film on the surface of the aluminium sheet avoids the direct exposure of the aluminium sheet to the SSC environment, which reduces the erosion of the aluminium sheet by the salt spray particles.” I wonder if you can provide an image or any kind of support for this argument.
Response: Thanks. Since aluminum is an oxygen-loving element, it can react with oxygen in the air at room temperature, and a dense film of Al2O3 will be formed, which can isolate the aluminum matrix from further corrosion. As described in reference [35-37] in the revised manuscript, the presence of oxide film enhances the corrosion resistance of the aluminum alloy. So, it can be summarized that the existence of the Al2O3 film on the surface of the aluminium sheet avoids the direct exposure of the aluminium sheet to the SSC environment, and reduces the erosion of the aluminium sheet by the salt spray particles. The relevant references are listed in the revised manuscript. (Page 6 of the revised manuscript, [35-37] of references).
- Please provide the chemical composition and mechanical properties of your alloys. What you have given in Tables 1 and 2 seems to be standard values. Please provide data from this specific (together with standard values).
Response: Thanks. The chemical composition and mechanical properties of studied alloys are added into the revised manuscript, and the standard and measured values are listed in Tables 1-2. (Pages 2-3 of the revised manuscript).
- While the fracture modes in Figures 6b and 6d (Figures 7b and 7d in the revised manuscript) are very well explained, there is hardly any explanation on the fracture surface, in Figure 6c (Figures 7c in the revised manuscript). In this figure, dimples are not observed. Please explain.
Response: Thanks. The crack of the aluminum plate extends from the area A of the aluminum plate (the area A is marked in Fig. 7a) to the area C, the crack extends along the inner wall of the aluminum alloy that adjoining the concave die. The crack extension direction is perpendicular to the load direction, so the obvious fracture characteristics can be seen in Figs. 7b and 7d. However, area B is the inner wall area of the aluminum alloy sheet that adjoining the concave die, and there is no intense tearing. The separation of the rivet from the lower aluminum plate causes the metal delamination phenomenon. Therefore, Therefore, unlike Figs. 7(b, d), no dimples are observed in Fig. 7(c). The corresponding explanation is added to the revised manuscript. (Page 8 of the revised manuscript).
- Please add scale bar to Figure 7a (Figures 8a in the revised manuscript).
Response: Thanks for your advice. Scale bar has been added in the revised manuscript (Page 9 of the revised manuscript).
- In Figure 7d (Figures 8d in the revised manuscript), what exactly are those spherical features on the fracture surface? Do you have any EDS analysis? If so, please add it to the manuscript.
Response: Thanks. From the morphology and formation position of the spherical substances on the fracture surface in Figure 8(d), as well as the previous work of the authors (references [21]), the spherical features are NaCl crystals that attached to the surface of the aluminum plate. Due to the long-time salt spray environment exposure, the salt spray particles will gradually erode into the joint gap, and the presence of corrosion pits also helps the attachment of salt spray particles. The corresponding reference is added to the revised manuscript. (Page 9 of the revised manuscript)
- Authors have argued that the fatigue micro-cracks initiation area is located at the junction of the lower aluminum sheet and the upper sheet and that this area becomes a vulnerable area due to the stress concentration caused by the corrosion pit. I wonder if you can provide images of mentioned pits.
Response: Thanks. A new section (3.3 Corrosion mechanism) has been added to the revised version manuscript. The corrosion appearance and corrosion pit locations have been marked in red in Fig. 3(b) (Page 4 of the revised manuscript).
- I strongly suggest authors give a short overview of different possible wear and corrosion mechanisms, maybe in a short paragraph. This is really missing in the manuscript. The current structure gives the feeling that authors first have introduced the problem and then sort of moved to the results. I think for readers it would be helpful to have a short overview of corrosion mechanisms perhaps somewhere at the end of introduction. I strongly suggest authors check and refer to the following articles. Though the topic might not be directly related to your article, but together they provide explanations for different corrosion mechanisms.
- Fan, B., Zhao, X., Liu, Z., Xiang, Y., & Zheng, X. (2022). Inter-component synergetic corrosion inhibition mechanism of Passiflora edulia Sims shell extract for mild steel in pickling solution: Experimental, DFT and reactive dynamics investigations. Sustainable Chemistry and Pharmacy, 29, 100821. doi: https://doi.org/10.1016/j.scp.2022.100821
- Xie, J., Zhang, J., Zhang, Z., Yang, Q., Guan, K., He, Y., Wu, R. (2022). New insights on the different corrosion mechanisms of Mg alloys with solute-enriched stacking faults or long period stacking ordered phase. Corrosion science, 198, 1. doi: 10.1016/j.corsci.2022.110163
- Gong, P., Wang, D., Zhang, C., Wang, Y., Jamili-Shirvan, Z., Yao, K., Wang, X., Corrosion behavior of TiZrHfBeCu(Ni) high-entropy bulk metallic glasses in 3.5 wt. % NaCl. npj Mater Degrad 6, 77 (2022). https://doi.org/10.1038/s41529-022-00287-5
Response: Thanks for your advice. In the SSC test the SPR specimens, pitting corrosion of 5052 aluminum alloy plate is the most common corrosion type. The relevant corrosion mechanism has been added to section 3.2 in the revised manuscript. Also, the above references are cited to support the corrosion mechanism analysis. (Page 4 of the revised manuscript, [29-31] and [33]).
- The conclusion is very short and does not fully reflect the findings of your article. Please present more details in your conclusion.
Response: Thanks for your advice. The second point of the conclusion section has been supplemented to make the conclusion more adequate. (Page 14 of the revised manuscript).

Reviewer 2 Report
The paper very thoroughly explains the influence of salt spray corrosion on the static (monotonic tension test) and dynamic (high cycle fatigue test) load capacity of the self-piercing riveted joint between titanium and aluminum sheets. At the beginning it is presented how the salt-spray corrosion duration influences the corrosion rate and then how the corrosion influences the load capacity by showing the tensile force-displacement (results of the monotonic tensile test) and F-N (results of the cyclic test) charts. There also is a simple equation presented that shows the relation between the maximal force, SSC duration and number of cycles to failure. Additionally, mechanisms of failures (static and dynamic) are explained in detail in relation to the SSC duration.
Although the paper is well written it reads more like a detailed test report on one simple case. There is some novelty in the paper but only to describe the setup (combination of the materials and geometry) used in the experiments. For the paper to be scientifically better (introduce high novelty) it would need to be more general. For instance, what would happen if the thickness of the sheets is increased or if the two plates are switched (aluminum on top and titanium on bottom). The problem is that presented results and all the findings are only valid for the specific case used in the research. For broader understanding some other factors like thickness of the sheets and position of the sheet (which one is on top and which one at the bottom in the connection) should be explained as well. Specially the later would be interesting to see since it might happen that installing the rivet from the other side would drastically change the load capacity which also could be shown by numerical simulation.
I also noticed that there is no explanation regarding the details of the cyclic test (what was the R ratio) which definitely should be included in the report.
Author Response
Ref: Materials-2066541
Title: Insight of salt spray corrosion on mechanical properties of TA1-Al5052 self-piercing riveted joint
Journal: Materials
Dear reviewers and editor:
Thanks for the suggestive advice from reviewers and editors. The manuscript has been carefully checked again. The following are the response to the comments from the reviewers and editor. The corresponding modifications have been made and the important corrections are highlighted in YELLOW in the revised manuscript.
Comments from the reviewers:
Reviewer 2#: The paper very thoroughly explains the influence of salt spray corrosion on the static (monotonic tension test) and dynamic (high cycle fatigue test) load capacity of the self-piercing riveted joint between titanium and aluminum sheets. At the beginning it is presented how the salt-spray corrosion duration influences the corrosion rate and then how the corrosion influences the load capacity by showing the tensile force-displacement (results of the monotonic tensile test) and F-N (results of the cyclic test) charts. There also is a simple equation presented that shows the relation between the maximal force, SSC duration and number of cycles to failure. Additionally, mechanisms of failures (static and dynamic) are explained in detail in relation to the SSC duration.
Although the paper is well written it reads more like a detailed test report on one simple case. There is some novelty in the paper but only to describe the setup (combination of the materials and geometry) used in the experiments. For the paper to be scientifically better (introduce high novelty) it would need to be more general. For instance, what would happen if the thickness of the sheets is increased or if the two plates are switched (aluminum on top and titanium on bottom). The problem is that presented results and all the findings are only valid for the specific case used in the research. For broader understanding some other factors like thickness of the sheets and position of the sheet (which one is on top and which one at the bottom in the connection) should be explained as well. Specially the later would be interesting to see since it might happen that installing the rivet from the other side would drastically change the load capacity which also could be shown by numerical simulation.
I also noticed that there is no explanation regarding the details of the cyclic test (what was the R ratio) which definitely should be included in the report.
Response: Thank you for your affirmation. According to the researches of scholars and the author's previous works, riveting parameters such as sheet thickness and sheet lap sequence have a great influence on the performance of self-pierce riveted joints. When different metal plates are connected, the plates with strong plastic deformation ability should be placed on the lower layer. At the same time, the rivet height should not exceed the total thickness of the plate by 3mm. An excellent joint should have strong mechanical properties and corrosion resistance. The purpose of this work is to investigate the effect of corrosion on the performance of the joint, so the selected parameters is adopted by the trial-and-error method. The related illustration has been added to section 2.2. Self-Piercing Riveting joints in the revised manuscript.
In addition, the stress ratio (R=1) for the fatigue test in this paper has been indicated in the revised manuscript. Thanks (Page 3 and page4 of the revised manuscript, [23-25] of references)

Round 2
Reviewer 1 Report
Authors have properly addressed all comments. The paper is acceptable in my opinion.
Reviewer 2 Report
In the revised version, the comments of the review have been properly taken into account and the purpose of the research has been additionally explained.
One typo is in page 8 (yellow text: 2x Therefore).